# Relationship between liver dysfunction, lipoprotein concentration and mortality during sepsis

Sébastien Tanaka[1,2☯]*, Christian De Tymowski[1,3,4☯], Jules Stern[1], Donia Bouzid[4,5,6], Nathalie Zappella[1], Aurélie Snauwaert[1], Tiphaine Robert[7], Brice Lortat-jacob[1], Alexy Tran-dinh[1,4,8], Pascal Augustin[1], Anne Boutten[7], Parvine Tashk[1], Katell Peoc'h[3,4,7], Olivier Meilhac[2,9], Philippe Montravers[1,4,10]

1 Assistance Publique—Hôpitaux de Paris (AP-HP), Department of Anesthesiology and Critical Care Medicine, Bichat-Claude Bernard Hospital, Paris, France, 2 Réunion Island University, French Institute of Health and Medical Research (INSERM), Diabetes atherothrombosis Réunion Indian Ocean (DéTROI), CYROI Plateform, Saint-Denis de La Réunion, Saint Denis, France, 3 French Institute of Health and Medical Research (INSERM), Center for Research on Inflammation, Paris, France, 4 Université de Paris, UFR Paris Nord, Paris, France, 5 Assistance Publique—Hôpitaux de Paris (AP-HP), Emergency Department, Bichat-Claude Bernard Hospital, Paris, France, 6 French Institute of Health and Medical Research (INSERM), Infection, Antimicrobials, Modelling, Evolution, Paris, France, 7 Assistance Publique—Hôpitaux de Paris (AP-HP), Biochemistry Department, Bichat-Claude Bernard Hospital, Paris, France, 8 French Institute of Health and Medical Research (INSERM), Laboratory for Vascular Translational Science, Paris France, 9 Réunion Island University-affiliated Hospital, Saint Denis, France, 10 French Institute of Health and Medical Research (INSERM), Physiopathology and Epidemiology of respiratory diseases, Paris, France

☯ These authors contributed equally to this work.
* sebastien.tanaka@aphp.fr

**Data Availability Statement:** All relevant data are within the paper and its Supporting Information files.

## Abstract

### Background

High-density lipoproteins (HDLs) are synthesized by the liver and display endothelioprotective properties, including anti-inflammatory, antiapoptotic, antithrombotic and antioxidant effects. In both septic and chronic liver failure patients, a low HDL cholesterol (HDL-C) concentration is associated with overmortality. Whereas sepsis-associated liver dysfunction is poorly defined, the aim of this study was to characterize the relationship between liver dysfunction, lipoprotein concentrations and mortality in septic patients in the intensive care unit (ICU).

### Methods

A prospective observational study was conducted in a university hospital ICU. All consecutive patients admitted for septic shock or sepsis were included. Total cholesterol, HDL-C, low-density lipoprotein-cholesterol (LDL-C), and triglyceride levels were assessed at admission. Sepsis-associated liver dysfunction was defined as a serum bilirubin≥ 2N or aspartate aminotransferase/alanine aminotransferase concentrations ≥ 2N. Short-term and one-year prognostic outcomes were prospectively assessed.

**Funding:** The authors received no specific funding for this work.

**Competing interests:** The authors have declared that no competing interests exist.

## Results

A total of 219 septic patients were included, and 15% of them presented with sepsis-associated liver dysfunction at admission. Low concentrations of lipoproteins were associated with mortality at Day 28 in the overall population. Sepsis-associated liver dysfunction at admission was associated with overmortality. In this subgroup, patients had a lower HDL-C concentration than patients without hepatic dysfunction (HDL-C = 0.31 [0.25, 0.55] mmol/L vs. 0.48 [0.29, 0.73] mmol/L, p = 0.0079) but there was no relationship with the outcome. Interestingly, no correlation was observed between lipoprotein concentrations and liver dysfunction markers.

## Conclusion

Sepsis-associated liver dysfunction at ICU admission is strongly associated with overmortality and is associated with a lower HDL-C concentration. However, in this subgroup of patients, HDL-C concentration had no relationship with mortality. Further exploratory studies are needed to better understand the interaction between lipoproteins and liver dysfunction during sepsis.

## Introduction

High-density lipoproteins (HDLs) are characterized by their ability to transport cholesterol from peripheral tissues back to the liver, which confers a major cardiovascular protective effect [1, 2]. In addition to this function, these nanoparticles display endothelial protective properties, including anti-inflammatory, antiapoptotic, antithrombotic and antioxidant effects, and can also bind and neutralize lipopolysaccharides (LPS) [3–5]. During inflammatory states and, in particular, during sepsis, many studies have shown both a significant decrease in the concentration of HDL cholesterol (HDL-C) in connection with overmortality [6–9] and a decrease in the functionality of HDL particles [10–14]. Supplementation with functional reconstituted HDLs or mimetic peptides has been tested in animal models of sepsis, and many studies have described positive effects on morbidity and mortality [15–18].

Mechanisms underlying the decrease in the concentration of HDL-C during sepsis are poorly described [4]. Several hypotheses have been proposed, such as the consumption of HDL particles, hemodilution, capillary leakage or decreased HDL synthesis by the liver, particularly in cases of sepsis-associated liver dysfunction [5, 19, 20].

Sepsis-associated liver dysfunction is not well documented, and its definition and pathophysiology are still controversial [21, 22]. Because HDL particles are synthetized by the liver and given that the reverse cholesterol transport (RCT) process is conditioned by hepatic pathways and receptors, liver dysfunction during sepsis could have an impact on HDL particles and HDL-C concentration.

Thus, the goals of the present study were:

- To assess sepsis-associated liver dysfunction in a cohort of septic patients in the intensive care unit (ICU),

- To compare lipoprotein and especially HDL-C concentration in septic patients with and without hepatic impairment at admission, and

- To investigate the relationship between sepsis-associated liver dysfunction—lipoprotein concentrations and the outcome.

## Materials and methods

### Study design

This was a prospective, observational monocentric study (HIGHSEPS cohort) conducted in the surgical ICU of Bichat-Claude Bernard University Hospital, Paris, France. The methods of this cohort have been previously published in a study involving 205 patients of the HIGHSEPS cohort [23]. All patients recruited from May 2016 to April 2020 admitted for septic shock or sepsis according to the criteria of the Surviving Sepsis Campaign were included [24]. All patients with preexisting liver disease, such as cirrhosis, fatty liver disease or liver cancer, and immunocompromised patients (acquired immune deficiency syndrome or transplant surgery) were excluded from the study. Patients with angiocholitis were also excluded from the analysis because the hepatic location of this sepsis could be a source of bias.

The study was approved by the French Society of Anesthesiology and Critical Care Medicine Research Ethics Board (HIGHSEPS study, IRB number 00010254). Written informed consent was obtained from participating patients.

Patient demographics, diagnosis, Simplified Acute Physiology Score II (SAPSII) and Sepsis-related Organ Failure Assessment (SOFA) severity scores [25, 26], organ supportive therapies, including renal replacement therapy and vasopressor use, and clinical data were prospectively collected. Data regarding the site of infection were gathered. Data on ICU and in-hospital mortality at 28 days (Day 28), duration of mechanical ventilation, number of days alive without mechanical ventilation at Day 28, length of stay in the ICU and hospital stay were collected. At admission, plasma concentrations of total cholesterol, HDL-C, LDL-C and triglycerides were measured. These analyses were performed in the Biochemistry Laboratory of Bichat Claude-Bernard Hospital. Total cholesterol (TC), HDL-C, TC/HDL-C ratio, LDL-C and triglyceride concentrations were determined by routine enzymatic assays (CHOL, HDLC and TRIG methods, Dimension VISTA® System, Siemens Healthineers™). The reference values for these assays were HDL-C >1.40 mmol/l, total cholesterol $4.40 < N < 5.20$ mmol/l and triglycerides $0.50 < N < 1.7$ mmol/l. According to the recommendations of the French National Authority for Health 2017 and the European Society of Cardiology 2016, LDL-C concentration targets have been established based on vascular risk factors [27].

The normal ranges of aspartate aminotransferase (ASAT) and alanine aminotransferase (ALAT) in our institution are [15–37] U/L and [16–63] U/L, respectively. The normal range of total bilirubin is <17 μmol/L.

### Definition

To our knowledge, there is no consensual definition of sepsis-associated liver dysfunction [28]. Many authors have used very different thresholds taking into account liver enzymes, total bilirubin or both [21, 22]. However, there is no consensus, including in international recommendations. Regarding total bilirubin threshold, some values have been extrapolated from recommendations concerning the management of hepatitis or drug-induced liver disorders [29–31]. Because there is no consensus and because hepatic impairment could have both ischemic and cholestatic etiology, we then decided to define hepatic dysfunction at admission as a total bilirubin $\geq 2N$ (i.e $\geq 34$ μmol/L) or ASAT and ALAT concentrations $\geq 2$ N.

### Statistical analysis

Qualitative data are expressed as absolute numbers and proportions and were compared using the chi-square test or Fisher's exact test as appropriate. Quantitative data are expressed as medians and interquartile ranges and were compared using the Mann–Whitney or Kruskal–

Wallis tests as appropriate. For Day-28 mortality discrimination, the receiver operating characteristic curve (ROC) analysis tested the best threshold values (using the Youden index) of total cholesterol (TC), HDL-C, TC/HDL-C ratio, LDL-C and triglycerides, and the area under the curve (AUC) was calculated. The Day-28 survival rate was analyzed by the Kaplan–Meier test and compared by a log rank test. To assess whether hepatic dysfunction was an independent factor of mortality, a multivariate analysis was carried out using a binary logistic regression model. Variables with nominal 2-tailed p values less than 0.1 were entered into the multivariate model, except for variables with obvious collinearity. The final models was selected using the backward stepwise regression using the AIC and the Tjur's R2 coefficient of discrimination.

All statistical analyses were performed using R software (R Core Team, 2014) figures were produced using 'ggplot2 package' and statistics using 'stat package'. A p value < 0.05 was considered statistically significant.

## Results

### Population

Two hundred twenty-six septic patients were prospectively and consecutively included in our ICU. Finally, 219 patients were included in the study (7 cases of angiocholitis were excluded from the analysis). Thirty-three (15%) patients were admitted with sepsis-associated liver dysfunction at ICU admission. In the sepsis-associated liver dysfunction subgroup, 27 (81%) patients had a total bilirubin ≥ 2N, and 16 (48%) patients had ASAT and ALAT concentrations ≥ 2 N.

Associated bacteriemia were also more numerous in sepsis-associated liver dysfunction patients (n = 14/33 (42%) vs. n = 47/186 (25%), p = 0.043). Admission SOFA and SAPSII scores were higher in patients with sepsis-associated liver dysfunction at admission (11 [9, 14] vs. 6 [4, 8], p <0.001 and 65 [56, 74] vs. 56 [40, 68], p = 0.002, respectively). Table 1 shows the general characteristics, etiology of sepsis and outcome of the patients.

### Relationship between sepsis-associated liver dysfunction and mortality

Sepsis-associated liver dysfunction was associated with 28-day mortality (see Table 1 and the Kaplan–Meier analysis in Fig 1).

In multivariate analysis, four factors were associated with 28-day mortality: sepsis-associated liver dysfunction, Odds-ratio (OR) = 2.44 CI [1.02, 5.75], p = 0.041; respiratory SOFA, OR = 1.40 CI [1.04, 1.90], p = 0.030; cardiovascular SOFA, OR = 1.50 CI [1.08, 2.33], p = 0.031 and age, OR = 1.03 CI [1.01, 1.06], p = 0.017.

### Relationship between lipoprotein concentration and hepatic dysfunction

At ICU admission, sepsis-associated liver dysfunction patients presented a significantly lower HDL-C concentration (HDL-C = 0.31 [0.25, 0.55] mmol/L vs. 0.48 [0.29, 0.73] mmol/L, p = 0.0079). No differences were found in TC, TG and LDL-C concentrations between groups. These results are shown in Fig 2. Interestingly, TC/HDL-C ratio was significantly higher in sepsis-associated liver dysfunction patients (TC/HDL-C = 7.0 [3.4, 9.7] vs. 4.5 [3.0, 7.0], p = 0.030).

### Relationship between lipoprotein concentrations at admission and mortality

To determine lipid and lipoprotein cutoff values to predict 28-day mortality, ROC curves were plotted, and the Youdden index was determined (S1 Fig). Fig 3 shows mortality at Day 28 as a

**Table 1. General characteristics of the population, outcome and type of sepsis, overall and according to sepsis-associated liver dysfunction.**

| Characteristic | Overall, N = 219[1] | No liver dysfunction N = 186 (85%)[1] | Liver dysfunction N = 33 (15%)[1] | p value[2] |
|---|---|---|---|---|
| Age (years) | 63 [52, 72] | 63 [52, 73] | 62 [49, 70] | 0.557 |
| Male | 117 (53) | 100 (54) | 17 (52) | 0.811 |
| Septic shock | 166 (76) | 136 (73) | 30 (91) | 0.028 |
| Associated-bacteriemia | 61 (28) | 47 (25) | 14 (42) | 0.043 |
| Peritonitis | 89 (41) | 74 (40) | 15 (45) | 0.541 |
| Urinary tract infections | 39 (18) | 34 (18) | 5 (15) | 0.665 |
| Skin and soft tissue infections | 36 (16) | 31 (17) | 5 (15) | 0.829 |
| Pleuro-pulmonary sepsis | 35 (16) | 29 (16) | 6 (18) | 0.708 |
| Other sepsis | 20 (9.1) | 18 (9.7) | 2 (6.1) | 0.745 |
| SAPS II at admission | 58 [43, 69] | 56 [40, 68] | 65 [56, 74] | 0.002 |
| SOFA at admission | 7.0 [4.0, 10] | 6.0 [4.0, 8.0] | 11.0 [9.0, 14.0] | <0.001 |
| Cardiovascular SOFA | 4.0 [1.0, 4.0] | 4.00 [1.00, 4.00] | 4.00 [4.00, 4.00] | 0.003 |
| Respiratory SOFA | 1.0 [0.0, 3.0] | 1.00 [0.00, 2.00] | 2.00 [1.00, 3.00] | <0.001 |
| Neurological SOFA | 0.0 [0.0, 1.0] | 0.00 [0.00, 1.00] | 0.00 [0.00, 1.00] | 0.137 |
| Coagulation SOFA | 0.0 [0.0, 1.0] | 0.00 [0.00, 0.00] | 0.00 [0.00, 2.00] | <0.001 |
| Kidney SOFA | 1.0 [0.0, 3.0] | 1.00 [0.00, 2.00] | 2.00 [1.00, 4.00] | <0.001 |
| Lactate at admission | 2.30 [1.50, 3.50] | 2.20 [1.40, 3.40] | 2.90 [2.10, 5.40] | 0.005 |
| Length of MV (days) | 2 [0, 8] | 2 [0, 6] | 6 [3, 8] | <0.001 |
| ICU length of stay (days) | 7 [3, 15] | 7 [3, 15] | 7 [4, 18] | 0.705 |
| ICU mortality | 42 (19) | 30 (16) | 12 (36) | 0.007 |
| Day-28 mortality | 42 (19) | 29 (16) | 13 (39) | 0.001 |
| Day-90 mortality | 55 (25) | 42 (23) | 13 (39) | 0.042 |
| One year mortality | 67 (32) | 52 (29) | 15 (47) | 0.046 |

[1] Median [IQR]; n (%)

[2] Wilcoxon rank sum test; Pearson's chi-squared test; Fisher's exact test

Continuous variables are expressed as the median and interquartile range (IQR) and were compared using the Mann–Whitney U test. Categorical variables are expressed as n (%) and were compared with Fisher's exact test. ICU, intensive care unit; MV, mechanical ventilation; SAPS II: Simplified Acute Physiology Score II; SOFA: Sepsis-related Organ Failure Assessment.

function of total cholesterol, triglyceride, HDL-C and LDL-C concentrations. Mortality at Day 28 of patients with total cholesterol concentrations <1.94 mmol/l at admission was significantly higher (log rank test, p<0.0001). Mortality at Day 28 of patients with TG concentration levels <0.87 mmol/l at admission was significantly higher (log rank test, p = 0.0012). Mortality at Day 28 of patients with HDL-C concentration levels below 0.33 mmol/l at admission was significantly higher (log rank test, p = 0.038). Mortality at Day 28 of patients with LDL-C concentrations less than 0.73 mmol/l at admission was significantly higher (log rank test, p<0.0001).

Interestingly, TC/HDL-C ratio at admission failed to discriminate patient mortality at Day 28 (chosen cut-off: TC/HDL-C = 6.5, log-rank test p = 0.59).

Moreover, in the subgroup of patients without liver dysfunction at admission, TC, TG, HDL-C and LDL-C concentrations at admission could stratify patients according to mortality at Day 28 (Fig 4). Mortality at Day 28 of patients with total cholesterol concentrations <1.94 mmol/l at admission was significantly higher (log rank test, p<0.0001). Mortality at Day 28 of patients with TG concentration levels <0.87 mmol/l at admission was significantly higher (log rank test, p = 0.0013). Mortality at Day 28 of patients with LDL-C concentrations <0.73 mmol/l at admission was significantly higher (log rank test, p<0.0001). Mortality at Day 28 of

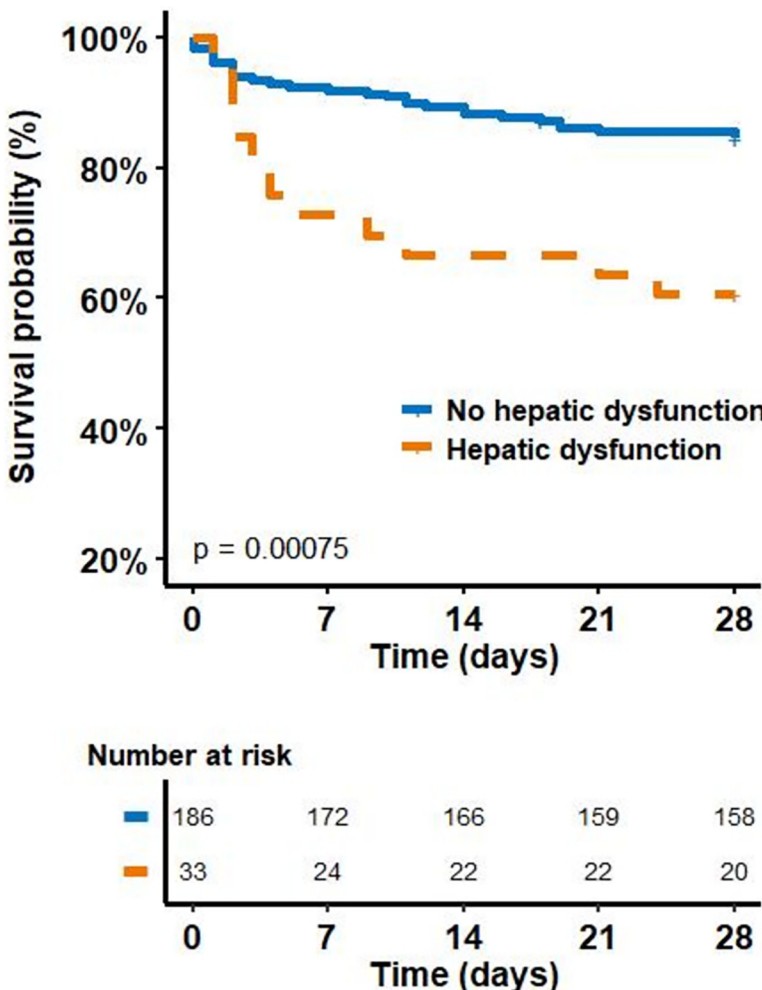

**Fig 1. Kaplan–Meier estimates of survival in the 28 days after the onset of sepsis for patients according to hepatic dysfunction status.**

patients with HDL-C concentration levels <0.33 mmol/l at admission was higher, but the difference was not statistically significant (log rank test, p = 0.056).

Interestingly, in the subgroup of patients with hepatic dysfunction, TC, TG, HDL-C and LDL-C concentrations at admission failed to discriminate patient mortality at Day 28 (log-rank test p = 0.15, p = 0.31, p = 0.97 and p = 0.62, respectively). These results are presented in Fig 4.

## Correlation between lipoprotein concentrations and hepatic dysfunction markers

Correlations between TC, TG, HDL-C, LDL-C and hepatic dysfunction markers are shown in Fig 5. Variations in lipid and lipoprotein values were mostly independent of other variations, especially of liver function tests. The only significant correlations were the association between LDL-C and total cholesterol (rho = 0.75) and triglycerides and HDL-C (rho = -0.41).

## Discussion

Sepsis-associated liver dysfunction is not well documented, nor is its prognosis. The definition and pathophysiology of this hepatic dysfunction also seem controversial [21, 22]. In a

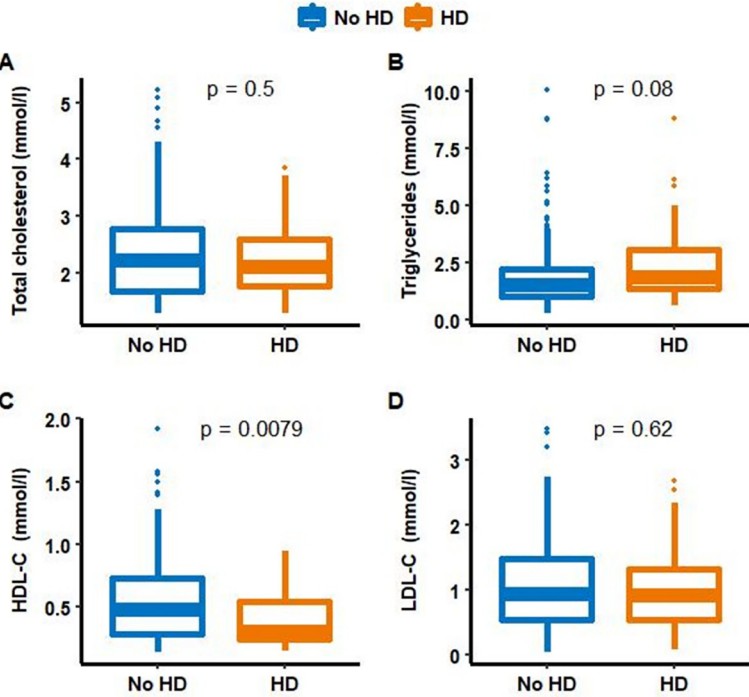

**Fig 2. Comparison between total cholesterol, high-density lipoprotein-cholesterol, low-density lipoprotein-cholesterol and triglyceride concentrations at admission, according to hepatic dysfunction status.** TC: total cholesterol, HDL-C: high-density lipoprotein-cholesterol, LDL: low-density lipoprotein-cholesterol, TG: triglycerides. HD: hepatic dysfunction.

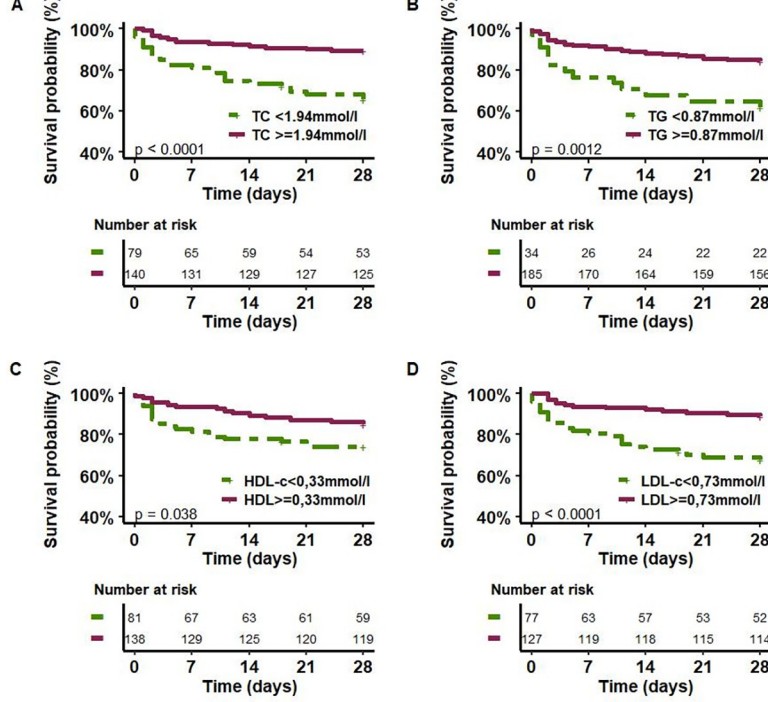

**Fig 3. Kaplan–Meier estimates of survival in the 28 days after the onset of sepsis for patients with different initial levels of lipoproteins.** TC: total cholesterol, HDL-C: high-density lipoprotein-cholesterol, LDL: low-density lipoprotein-cholesterol, TG: triglycerides.

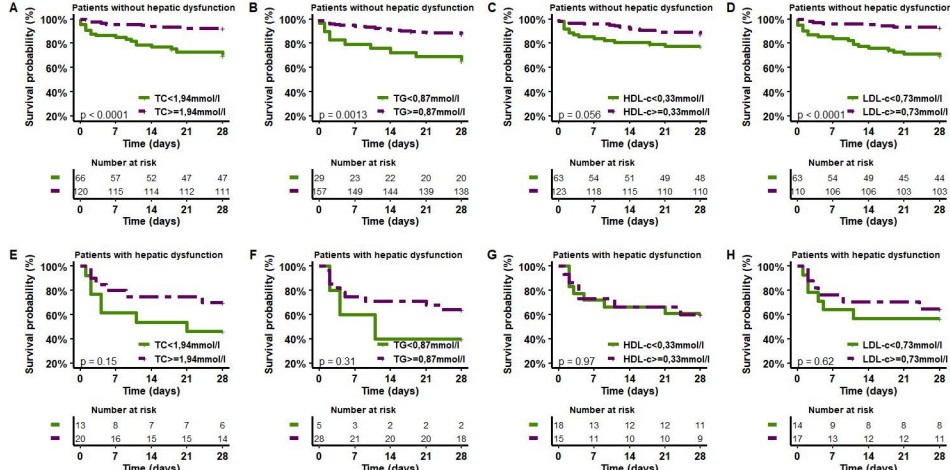

**Fig 4. Kaplan–Meier estimates of survival in the 28 days after the onset of sepsis for patients with different initial levels of lipoproteins, according to hepatic dysfunction status.** TC: total cholesterol, HDL-C: high-density lipoprotein-cholesterol, LDL: low-density lipoprotein-cholesterol, TG: triglycerides.

prospective study of 608 patients with sepsis stratified according to cholestasis at admission [32], those with cholestasis had higher severity scores and mortality than those without cholestasis. Kobashi et al. reported sepsis-associated liver dysfunction in 34.7% of the patients and distinguished 3 groups: "hepatocellular" (21.8%), "cholestatic" (48.1%) and "shock liver"

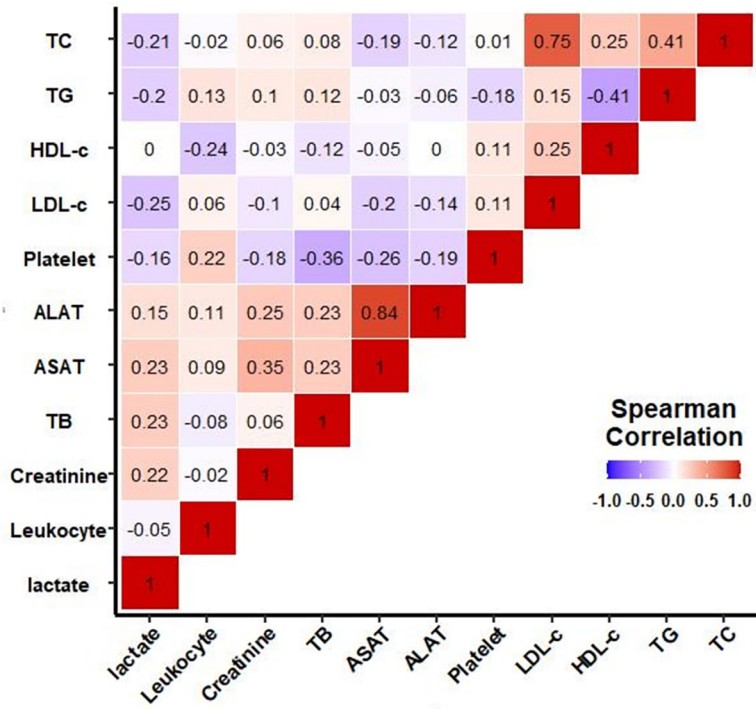

**Fig 5. Correlation between lipoprotein, triglycerides, total cholesterol and hepatic markers.** TC: total cholesterol, HDL-C: high-density lipoprotein-cholesterol, LDL: low-density lipoprotein-cholesterol, TG: triglycerides, TB: total bilirubin, ASAT: aspartate aminotransferase, ALAT: alanine aminotransferase.

(30.1%) [33]. As the incidence of sepsis-associated liver dysfunction is difficult to establish due to the lack of a homogeneous definition, we decided to base our definition of liver dysfunction both on the plasma bilirubin value and on liver enzyme concentrations. Hepatic dysfunction can be caused either by a cholestatic mechanism or by cytolysis or sometimes both. As there is no definition retained, we considered that using bilirubin value and liver enzyme concentration in the definition seemed the most rational choice.

We have shown in a previous study of the HIGHSEPS cohort a strong association between lipoprotein concentrations and mortality during sepsis [23]. In this same cohort, to the best of our knowledge, this is the first study to investigate the relationship between lipoproteins and sepsis-associated liver dysfunction.

The relationship between lipoproteins (especially HDL-C) and sepsis is well documented, with a significant decrease in lipoprotein concentrations during septic states [6–8, 23]. However, the mechanisms underlying the decreased HDL-C concentration during sepsis are poorly described [5, 20]. In our study, we showed that the HDL-C concentration was lower in patients with liver dysfunction than in those without liver dysfunction. This finding allows us to make a number of assertions. First, a decrease in the hepatic synthesis of HDL particles may account for this decrease. However, the absence of a difference between the two groups concerning the concentration of LDL does not support this theory because LDL particles are also produced by the liver. Second, the interaction between the decreased plasma HDL-C concentration and hepatic dysfunction may be due to an increased hepatic elimination of HDL particles in the context of bacterial sepsis. Indeed, HDL particles were shown to bind and promote lipopolysaccharide and/or bacterial clearance via the liver and subsequent bile excretion [34]. A recent study conducted by our team supports this theory [15]: after intra-abdominal injection of indium-labeled bacteria in mice, followed by an intravenous injection of reconstituted human HDL particles, the scintigraphic study showed an initial signal in the intra-abdominal region and then in the gall bladder. Finally, hepatic dysfunction could have consequences on reverse cholesterol transport (RCT) via the hepatic SRB1 receptor. This receptor allows the elimination of cholesterol by the liver and the initiation of a new cycle of RCT [35]. Thus, the hypothesis of an upregulation of SRB1 in the case of liver dysfunction could support the greater decrease in HDL-C in patients with sepsis-associated liver dysfunction relative to other septic patients. Nevertheless, this hypothesis needs confirmation.

An interesting point of our study is the lack of a relationship between HDL-C concentration and mortality in patients with sepsis-associated liver dysfunction. First, the small number of patients with this dysfunction may not be sufficient to unveil a potential link with mortality. Second, although the literature is controversial regarding the link between mortality and sepsis-associated liver dysfunction, a protective effect of HDL on this dysfunction can be discussed. Indeed, sepsis leads to not only quantitative but also qualitative modifications of HDL particles, including size and structural modifications [10–12, 36]. For example, proteomic studies in inflammatory states, such as sepsis, have demonstrated major changes in the composition of these particles, particularly in inflammatory proteins, such as a replacement of apoA1 by serum amyloid A (SAA), which may have an impact on patient outcomes [37–39]. This proinflammatory shift has been shown both during bacterial sepsis and during sepsis related to COVID-19 pneumonia [40]. Given that SAA is synthesized by the liver, an alteration of the hepatic production of this protein could have consequences on the functionality of these HDL particles and, thus, could lead to a protective effect. Third, the upregulation of the RSB1 receptor can lead to greater elimination of HDL particles and thus increase the clearance of LPS (for Gram-negative bacteria) and lipoteichoic acid (for Gram-positive bacteria) and thus have an impact on the outcome. This hypothesis nevertheless deserves to be verified in pre-clinical studies. Finally, in contrast with chronic liver failure, where HDL seems to be a robust

predictor of survival [41], the lack of a relationship between lipoprotein concentration and mortality in the case of sepsis-associated liver dysfunction reflects the fact that sepsis is a complex entity in which several organ dysfunctions are intertwined.

Our study has several limitations. First, it was a monocentric study conducted in a surgical ICU with a majority of patients with abdominal sepsis that could increase the proportion of hepatic dysfunction. Second, because of the nature of this prospective cohort study, the number of patients necessary to reach statistical significance was not calculated a priori. Ultimately, the low proportion of hepatic dysfunction in our cohort would merit increasing the number of patients in the cohort. Third, inflammatory parameters, such as cytokines, were not analyzed in our study. Finally, in the absence of a clear consensus, we chose a definition of liver dysfunction that took into account cytolysis and cholestasis, which may be controversial and deserves to be explored in the future.

## Conclusion

Sepsis-associated liver dysfunction at ICU admission is associated with overmortality at Day 28 and is associated with a lower HDL-C concentration. However, in this subgroup of hepatic dysfunction, HDL-C concentration at admission had no relationship with mortality. This study also found no correlation between lipoprotein and especially HDL-C concentration and hepatic dysfunction markers, such as ASAT, ALAT or lactate concentrations, reflecting the complexity of this sepsis-associated liver dysfunction entity. Further experimental mechanistic studies are necessary to better characterize the relationship between lipoproteins and hepatic dysfunction. It would also be interesting to look at HDL dysfunctions in patients with chronic liver disease in acute liver failure complicated by septic shock. In addition, high-powered studies are necessary to better define sepsis-associated liver dysfunction.

## Supporting information

**S1 Fig. Total cholesterol, triglycerides, HDL-C, LDL-C ROC cutoff values to predict 28-day mortality.** ROC curves were plotted to determine lipid and lipoprotein cutoff values to predict 28-day mortality. T-cholesterol: Total cholesterol; TG: triglycerides; HDL-C: high-density lipoprotein cholesterol; LDL-C: low-density lipoprotein cholesterol.
(TIF)

**S1 File.**
(XLSX)

## Acknowledgments

We would like to thank the medical and paramedical team of the Bichat Claude Bernard Surgical ICU, Paris, France.

## Author Contributions

**Conceptualization:** Sébastien Tanaka, Christian De Tymowski, Donia Bouzid, Olivier Meilhac, Philippe Montravers.

**Data curation:** Sébastien Tanaka, Christian De Tymowski, Jules Stern, Donia Bouzid, Nathalie Zappella, Aurélie Snauwaert, Tiphaine Robert, Brice Lortat-jacob, Alexy Tran-dinh, Pascal Augustin, Anne Boutten, Parvine Tashk, Katell Peoc'h, Olivier Meilhac.

**Formal analysis:** Sébastien Tanaka, Donia Bouzid, Aurélie Snauwaert, Katell Peoc'h, Philippe Montravers.

**Investigation:** Sébastien Tanaka, Nathalie Zappella, Aurélie Snauwaert, Tiphaine Robert, Brice Lortat-jacob, Alexy Tran-dinh, Pascal Augustin, Anne Boutten, Parvine Tashk, Katell Peoc'h, Philippe Montravers.

**Methodology:** Sébastien Tanaka, Christian De Tymowski, Donia Bouzid, Philippe Montravers.

**Resources:** Sébastien Tanaka, Jules Stern, Tiphaine Robert, Philippe Montravers.

**Software:** Christian De Tymowski.

**Supervision:** Sébastien Tanaka, Jules Stern, Nathalie Zappella, Philippe Montravers.

**Validation:** Sébastien Tanaka, Nathalie Zappella, Aurélie Snauwaert, Tiphaine Robert, Brice Lortat-jacob, Alexy Tran-dinh, Pascal Augustin, Anne Boutten, Parvine Tashk, Katell Peoc'h, Olivier Meilhac, Philippe Montravers.

**Visualization:** Sébastien Tanaka, Jules Stern, Olivier Meilhac, Philippe Montravers.

**Writing – original draft:** Sébastien Tanaka, Christian De Tymowski, Philippe Montravers.

**Writing – review & editing:** Olivier Meilhac, Philippe Montravers.

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
