## [Decision Letter · Decision Letter 0]

1 Jun 2022

PONE-D-22-09335Relationship between liver dysfunction and lipoprotein concentration in septic ICU patients Short-title: lipoprotein and liver during sepsisPLOS ONE

Dear Dr. TANAKA,

Thank you for submitting your manuscript to PLOS ONE. After careful consideration, we feel that it has merit but does not fully meet PLOS ONE’s publication criteria as it currently stands. Therefore, we invite you to submit a revised version of the manuscript that addresses the points raised during the review process.

We look forward to receiving your revised manuscript.

Kind regards,

Chiara Lazzeri

Academic Editor

PLOS ONE

Journal Requirements:

5. One of the noted authors is a group or consortium “Philippe MONTRAVERS”.In addition to naming the author group, please list the individual authors and affiliations within this group in the acknowledgments section of your manuscript. Please also indicate clearly a lead author for this group along with a contact email address.

Reviewers' comments:

Reviewer's Responses to Questions

**Comments to the Author**

1. Is the manuscript technically sound, and do the data support the conclusions?

Reviewer #1: Partly

Reviewer #2: Yes

2. Has the statistical analysis been performed appropriately and rigorously? 

Reviewer #1: No

Reviewer #2: Yes

3. Have the authors made all data underlying the findings in their manuscript fully available?

Reviewer #1: Yes

Reviewer #2: Yes

4. Is the manuscript presented in an intelligible fashion and written in standard English?

Reviewer #1: Yes

Reviewer #2: Yes

5. Review Comments to the Author

Reviewer #1: This mono-centric and non randomized study intended to correlate, in sepsis and septic shock ICU patients,Sepsis Associated Liver Dysfunction (SALD) and lipoprotein levels ( mostly HDL-C) at admission. The influence of those two settings upon mortality was carefully studied. The aim of this study was to better understand SALD, which is frequent during severe infections, and the dynamic of HDL-C, known to be decreased during this syndrome, and related with mortality. HDL-C has multiple positive effects upon inflammation, thrombosis, apoptosis and oxydation, which are all profoundly altered during sepsis.

Sepsis is still a very complex syndrome and any new data will help understanding the many mechanisms leading to it.

In this respect, the study/paper proposed for publication in the journal is interesting.

However, I have several serious concerns as well as minor ones.

SERIOUS CONCERNS

1) One of the most important one is the poor definition of SALD in the literature, and consequently in this paper.

For exemple, the HDL-C levels cut-off varie from 33 mmol/L ( which has been selected in the study, another one published in Critical Care in 2012) up to 70, or even more. It was 68,3 in the Surviving Sepsis Campaign studies.The aminiotransferase cut-off levels are more consensual ( twice N). The severity scores, like SAPS II or SOFA do not really help. In SAPS II, the bilirubin levels are used to calculate the score only in icteric patients. In the SOFA score, only bilirubin levels are considered, and, if we take the "liver part" of the score, 33mmol/L is just between score 1 and 2h.

Even if we can understand that it has been a nightmare to define SALD in this study, I do think that it's definition i s somewhat arbitrary.

2) The difference in mortality between patients with or without SALD is significant in the mono-variate analysis, but there are many differences between the two groups, in particular for severity indexes ( SAPS 2, SOFA, lactate). I might be wrong, but I have nor seen the results of a multi-variate analysis demonstrating that mortality is higher in the patients with SALD. It is important to see, and to discuss that there is no difference in mortality at Day 90, and 1 year

3) The assertions presented at the beginning of the discussion, which is unusual, are not very easy to understand.

The assertion 2 says that low HDL-C are correlated with mortality in patients with, or without SALD.

On line 321, it is said, at the opposite, that there is no correlation between HDL-C and mortality in patients with SALD. It is said again in the conclusion, line 356.

4) The lack of correlation between lipoprotein levels and mortality noted only in SALD patients is surprising, and very interesting, at least for me. The discussion on this finding ( 336 to 339) is very poor

5) As mentioned by the authors, the number of patients with SALD is rather low, and might explain some of the

strange findings.

MINOR CONCERNS

- May-be, you could add the word "mortality" in the title

- From line 288 to 290: the text is difficult to understand

- I think there are too many figures, in particular Kaplan Meir curves.

- The words "septic patients" and "severe sepsis" do not exist anymore in the last consensus documents

- I do think that angiocholitis patients must be exclusion cases

- Line 250, I do not understand the word "slightly similar"

-

Reviewer #2: Metabolic abnormalities are one of the important causes of death in sepsis. The liver is an important organ of body metabolism. It is of great clinical significance to study the correlation between liver lipid metabolism and the prognosis of sepsis. The article carried out long-term follow-up of 226 sepsis patients, which is a very excellent clinical study. Main comments: The data is huge, and further data mining should be carried out. Such as: whether there is a correlation between the ratio of apolipoprotein and prognosis.

6. PLOS authors have the option to publish the peer review history of their article (what does this mean?). If published, this will include your full peer review and any attached files.

Reviewer #1: No

Reviewer #2: **Yes: **Yin Yongjie

---

## [Author Response · Author response to Decision Letter 0]

11 Jul 2022

Dear Editor, 

Thank you for reviewing our manuscript entitled “Relationship between liver dysfunction and lipoprotein concentration in septic ICU patients”.

We appreciate reviewers’ comments and relevant questions that will help to improve the manuscript. 

Please find attached a revised version of our manuscript and a point-by-point response to the reviewer comments. 

We hope that our manuscript will now be suitable for publication in Plos One.

Reviewer #1: This mono-centric and non randomized study intended to correlate, in sepsis and septic shock ICU patients,Sepsis Associated Liver Dysfunction (SALD) and lipoprotein levels ( mostly HDL-C) at admission. The influence of those two settings upon mortality was carefully studied. The aim of this study was to better understand SALD, which is frequent during severe infections, and the dynamic of HDL-C, known to be decreased during this syndrome, and related with mortality. HDL-C has multiple positive effects upon inflammation, thrombosis, apoptosis and oxydation, which are all profoundly altered during sepsis.

Sepsis is still a very complex syndrome and any new data will help understanding the many mechanisms leading to it.

In this respect, the study/paper proposed for publication in the journal is interesting.

However, I have several serious concerns as well as minor ones.

SERIOUS CONCERNS

1) One of the most important one is the poor definition of SALD in the literature, and consequently in this paper.

For exemple, the HDL-C levels cut-off varie from 33 mmol/L ( which has been selected in the study, another one published in Critical Care in 2012) up to 70, or even more. It was 68,3 in the Surviving Sepsis Campaign studies. The aminiotransferase cut-off levels are more consensual ( twice N). The severity scores, like SAPS II or SOFA do not really help. In SAPS II, the bilirubin levels are used to calculate the score only in icteric patients. In the SOFA score, only bilirubin levels are considered, and, if we take the "liver part" of the score, 33mmol/L is just between score 1 and 2h.

Even if we can understand that it has been a nightmare to define SALD in this study, I do think that it's definition i s somewhat arbitrary.

Thank you for this very important comment and the word nightmare is not an understatement. We agree with the reviewer, our definition may seem arbitrary, but it seemed important to us that the two mechanisms (cholestasis and cytolysis) be included together in the definition because these mechanisms are really very intricate. 

Furthermore, we fully understand that the choice of the bilirubin threshold is complicated in the light of the literature. Nevertheless, several authors (including us in previous studies) have extrapolated data from studies and recommendations on hepatitis or drug-induced liver disorders (Bénichou et al J of hepatology 1990 PMID: 2254635 ; De Tymowski et al. J of hepatology 2019 PMID: 31152758 ; Wendel Garcia et al Critical care 2022 PMID: 35606831). And in light of these studies, the threshold of 34 µmol/l (i.e. 2N) seems to be relevant. In this context, we decided to take this threshold into consideration in our definition. As no patient in our cohort has bilirubin between 32 and 34 µmol/l, no change was performed. 

We thus have modified our definition in the text and we have added additional bibliographic references: “Many authors have used very different thresholds taking into account liver enzymes, total bilirubin or both. However, there is no consensus, including in international recommendations. Regarding total bilirubin threshold, some values have been extrapolated from recommendations concerning the management of hepatitis or drug-induced liver disorders. Because there is no consensus and because hepatic impairment could have both ischemic and cholestatic etiologies, we then decided to define hepatic dysfunction at admission as a total bilirubin ≥ 2N (i.e ≥ 34 µmol/L) or ASAT and ALAT concentrations ≥ 2N”. 

2) The difference in mortality between patients with or without SALD is significant in the mono-variate analysis, but there are many differences between the two groups, in particular for severity indexes ( SAPS 2, SOFA, lactate). I might be wrong, but I have nor seen the results of a multi-variate analysis demonstrating that mortality is higher in the patients with SALD. 

Thank you for this very important remark. To complete the analysis, we performed a multivariate analysis from the univariate analysis and thus 5 factors are associated with 28-day mortality including SALD : SALD, Odds-ratio (OR) = 2.44 CI [1.02, 5.75], p = 0.041 ; respiratory SOFA, OR = 1.40 CI [1.04, 1.90], p = 0.030 ; cardiovascular SOFA, OR = 1.50 CI [1.08, 2.33], p = 0.031 and age, OR = 1.03 CI [1.01, 1.06], p = 0.017. We added these important results in the analysis. 

 It is important to see, and to discuss that there is no difference in mortality at Day 90, and 1 year. 

As requested by the reviewer, we removed angiocholitis from our analysis (minor concerns part). The new results show an association between SALD and 28-day, 90-day and 1 year mortality (see Table 1). 

3) The assertions presented at the beginning of the discussion, which is unusual, are not very easy to understand.

We apologize for the lack of clarity at the beginning of the discussion. We wanted to summarize the important points of the results section but following your comments, it seemed to us more appropriate to withdraw this paragraph in the discussion section.

The assertion 2 says that low HDL-C are correlated with mortality in patients with, or without SALD.

We apologize for the lack of clarity of this sentence. There is a correlation with mortality when we take the whole cohort and in the subgroup of patients without hepatic dysfunction. As mentioned in the answer of the question 3, we removed this sentence of the manuscript. 

On line 321, it is said, at the opposite, that there is no correlation between HDL-C and mortality in patients with SALD. It is said again in the conclusion, line 356.

These sentences are correct. 

4) The lack of correlation between lipoprotein levels and mortality noted only in SALD patients is surprising, and very interesting, at least for me. The discussion on this finding (336 to 339) is very poor.

We apologize if our discussion on this subject (line 320 à 339) has not been extensive enough: “First, the small number of patients with this dysfunction may not be sufficient to unveil a potential link with mortality. Second, although the literature is controversial regarding the link between mortality and sepsis-associated liver dysfunction, a protective effect of HDL on this dysfunction can be discussed. Indeed, sepsis leads to not only quantitative but also qualitative modifications of HDL particles, including size and structural modifications (10–12,33). For example, proteomic studies in inflammatory states, such as sepsis, have demonstrated major changes in the composition of these particles, particularly in inflammatory proteins, such as a replacement of apoA1 by serum amyloid A (SAA), which may have an impact on patient outcomes (34–36). This proinflammatory shift has been shown both during bacterial sepsis and during sepsis related to COVID-19 pneumonia (37). Given that SAA is synthesized by the liver, an alteration of the hepatic production of this protein could have consequences on the functionality of these HDL particles and, thus, could lead to a protective effect. Finally, in contrast with chronic liver failure, where HDL seems to be a robust predictor of survival (38), the lack of a relationship between lipoprotein concentration and mortality in the case of sepsis-associated liver dysfunction reflects the fact that sepsis is a complex entity in which several organ dysfunctions are intertwined”. 

Nevertheless, to explain this observation, another hypothesis can be added: the upregulation of the RSB1 receptor can lead to greater elimination of HDL particles and thus increase the clearance of LPS (for Gram-negative bacteria) and lipoteichoic acid (for Gram-positive bacteria) and thus have an impact on the outcome. This hypothesis nevertheless deserves to be verified in pre-clinical studies. We added this point in the discussion.

5) As mentioned by the authors, the number of patients with SALD is rather low, and might explain some of the strange findings.

This remark is very relevant. As our study was only exploratory, our results are undeniably underpowered. It would be interesting to increase the number of patients in order to ultimately increase the number of patients with liver dysfunction. We then have re-emphasized this point in the limitations of this study at the end of the discussion: “Second, because of the nature of this prospective cohort study, the number of patients necessary to reach statistical significance was not calculated a priori. Ultimately, the low proportion of hepatic dysfunction in our cohort would merit increasing the number of patients in the cohort”. 

MINOR CONCERNS

- May-be, you could add the word "mortality" in the title

Reviewer 1 is totally right, this word must be present in the title. We then changed for “Relationship between liver dysfunction, lipoprotein concentration and mortality during sepsis”.

- From line 288 to 290: the text is difficult to understand

We are well aware that the sentence is not clear. We wanted to indicate to the reader that hepatic dysfunction could have a cholestatic cause or a cytolytic cause or both. In this context, we have modified the sentence “We, thus, considered that the cholestatic part and the cytolytic part had as much their role in hepatic dysfunction, but this assertion needs confirmation” by the followings: “Hepatic dysfunction can be caused either by a cholestatic mechanism or by cytolysis or sometimes both. As there is no definition retained, we considered that using bilirubin value and liver enzyme concentration in the definition seemed the most rational choice”. 

- I think there are too many figures, in particular Kaplan Meir curves.

Thank you for this comment. We have therefore decided to withdraw the Kaplan Meier curves from patients in septic shock. The additional figure has therefore been removed from the manuscript.

- The words "septic patients" and "severe sepsis" do not exist anymore in the last consensus documents

Reviewer 1 is totally right, we changed for sepsis and septic shock in the text. 

- I do think that angiocholitis patients must be exclusion cases

As requested by reviewer 1, we removed angiocholitis patients. Removing them is very relevant since sepsis whose origin is hepatic, may induce obvious biases regarding the presence or absence of hepatic dysfunction. The results from the new analyses show the same overall results. We changes all new data in the text, Tables and Figures. 

- Line 250, I do not understand the word "slightly similar"

We understand that this sentence is difficult to understand. We just wanted to explain that the results between sepsis and septic shock patients did not differ. In order to limit the number of figures, we have decided to remove this part of the manuscript.

Reviewer #2: Metabolic abnormalities are one of the important causes of death in sepsis. The liver is an important organ of body metabolism. It is of great clinical significance to study the correlation between liver lipid metabolism and the prognosis of sepsis. The article carried out long-term follow-up of 226 sepsis patients, which is a very excellent clinical study. Main comments: The data is huge, and further data mining should be carried out. Such as: whether there is a correlation between the ratio of apolipoprotein and prognosis.

We thank the reviewer for this kind comment. 

Thanks for suggesting to add some data like ratios. Indeed, some ratios can be very informative, especially in cardiovascular pathologies, but have not been studied extensively in sepsis. The most relevant ratio is probably the TC/HDL-C ratio. Interestingly, this ratio was higher in sepsis-associated liver dysfunction patients (TC/HDL-C = 7.0 [3.4, 9.7] vs. 4.5 [3.0, 7.0], p= 0.030). We added this point in the result section. 

Moreover, TC/HDL-C ratio at admission failed to discriminate patient mortality at Day 28 (log-rank test p=0.59).We also added this new information in the result section. 

Journal Requirements:

We have modified the manuscript according to Plos One’s style. 

We added in the manuscript that written informed consent was obtained from participating patients.

Data are available from the authors upon reasonable request.

We have removed the ethics statement at the end of the manuscript. Henceforth this statement is only to be found in the method part of this manuscript.

5. One of the noted authors is a group or consortium “Philippe MONTRAVERS”.In addition to naming the author group, please list the individual authors and affiliations within this group in the acknowledgments section of your manuscript. Please also indicate clearly a lead author for this group along with a contact email address.

It is a mistake, Philippe Montravers is not a consortium but the last author of the manuscript.

---

## [Editor Report · Decision Letter 1]

19 Jul 2022

Relationship between liver dysfunction, lipoprotein concentration and mortality during sepsis

PONE-D-22-09335R1

Dear Dr. TANAKA,

We’re pleased to inform you that your manuscript has been judged scientifically suitable for publication and will be formally accepted for publication once it meets all outstanding technical requirements.

Kind regards,

Chiara Lazzeri

Academic Editor

PLOS ONE
---

## [Editor Report · Acceptance letter]

11 Aug 2022

PONE-D-22-09335R1 

Relationship between liver dysfunction, lipoprotein concentration and mortality during sepsis 

Dear Dr. Tanaka:

I'm pleased to inform you that your manuscript has been deemed suitable for publication in PLOS ONE. Congratulations! Your manuscript is now with our production department. 

Kind regards, 

on behalf of

Dr. Chiara Lazzeri 

Academic Editor

PLOS ONE